# Compositions of Abrasive Cosmetics from Polish Manufacturers

Martyna Guzik [1], Olga Czerwińska-Ledwig [2] and Anna Piotrowska [2,*]

1 Scientific Club at the Department of Chemistry and Biochemistry, University of Physical Education, 31-571 Krakow, Poland
2 Institute for Basic Sciences, Faculty of Physiotherapy, University of Physical Education, 31-571 Krakow, Poland
* Correspondence: anna.piotrowska@awf.krakow.pl

**Abstract:** Microplastics have been widely used in cosmetics and, among other things, very often as an abrasive component in peelings. This type of additive is not the main cause of environmental microplastic contamination, but it can pose a significant threat to the environment and to people. Manufacturers are increasingly taking the decision to withdraw microplastics from cosmetics, replacing them with alternatives, and this is also happening because of legal requirements. The European Chemicals Agency, in 2019, presented a proposal to limit the use of polymer plastics in cosmetic products due to the fact that they may be a potential source of primary microplastics. The final form of the EU regulation is planned for the years 2023–2024. The aim of this study was to analyze the compositions of widely available rinse-off abrasive cosmetics from Polish manufacturers and to identify the most common natural raw materials replacing microplastics. Fifty randomly selected rinse-off products were analyzed for abrasive ingredients in INCI (International Nomenclature of Cosmetic Ingredients) formulations. Among the tested cosmetics, 13 contained microplastics and 49 contained natural abrasive particles, and polyethylene did not appear in any product. The most common vegetable raw material substitute for microplastics was sugar, and sodium chloride was the most common mineral substitute. Compared to previous years, there has been an improvement in the Polish cosmetics market, where manufacturers are increasingly opting for plant-based substitutes for microplastics, but relevant legal regulation is still needed.

**Keywords:** microplastic; abrasive cosmetics; cosmetic raw materials





## 1. Introduction

### 1.1. Reasons for Withdrawal of Microplastics from Abrasive Cosmetics

Pollution caused by the production of plastics is one of the most important environmental problems today. Annual global production of plastic materials is about 300 million tons. This value has increased more than 20 times in the last sixty years. Only 30% of plastic is recycled in Europe. About 10% of the plastic produced enters the oceans each year, while about 8 million pieces of plastic enter the oceans daily, mainly from waters carried by river currents. Plastics are highly durable, making it possible for them to take hundreds of years to decompose [1].

From the moment it is produced, the plastic is broken down into smaller particles. The mechanism may include mechanical and chemical decomposition (depolymerization), e.g., under the influence of sunlight, microbes and water. Plastic fragments that are less than 5 mm are defined as microplastics. Nanoplastics are particles that do not exceed <1 μm [1,2].

Plastic microparticles can be divided into primary and secondary types. The first ones are produced for a specific purpose and are found, for example, in the form of microspheres in personal care products, as plastic pellets used in industrial production or as plastic fibers in synthetic textiles. Secondary microplastics are formed from the degradation and

fragmentation of larger plastics and from the degradation of synthetic fibers. Nanoplastic is the product of further degradation of both mentioned microplastic types [1,3].

Nanoplastic particles present in the environment are characterized by irregular shape and size, heterogeneity of composition and a wide variety of physical properties. Nanoplastics are not neutral to human health. Daily exposure, most often in occupational environments, to air contaminated with plastic nanofibers can lead to chronic diseases of the respiratory system and other organs. Three routes can be distinguished for the entry of this material into the body. Inhalation is the first of these and is most often associated with the pollution of the air with aerosols that contain nanoplastics. The second pathway is transdermal absorption (through hair follicles, sweat glands or through damaged skin), which occurs through contact with contaminated air, with contaminated water, and when using cosmetics containing nanoplastics, such as abrasive cosmetics. The third route for nanoplastic penetration into the human body is through the gastrointestinal tract, where plastic nanoparticles most often enter with fish-meat, seafood and contaminated water [4].

In response to increasing pollution from polymer materials, the European Union has developed the concept of a circular economy for plastics, which implies that, once produced and put into circulation, products, raw materials and materials must remain in the system for as long as possible. This reduces the release of polymer plastics into the environment, as well as the production of new waste [5]. With concern about microplastics in the water environment, which can directly and indirectly affect human health, Directive 2020/2184 of the European Parliament and the Council (EU) was endorsed to improve the availability of safe drinking water [5]. The European Chemicals Agency (ECHA), in 2019, proposed restricting the use of polymeric plastics in cosmetic products, due to the fact that they can be a potential source of primary microplastics. The list in the ECHA document included 19 such compounds [6]:

- Polyethylene (INCI: Polyethylene): abrasive, film-forming, viscosity-regulating;
- Polypropylene (INCI: Polypropylene): viscosity-regulating;
- Polymethyl methacrylate (INCI: Polymethyl methacrylate): film-forming, sorbent to deliver active ingredients;
- Poly (tetrafluoroethylene) (INCI: Polytetrafluoroethylene acetoxypropyl betaine): improves hair condition, filler, slip enhancer, binding agent, improves skin condition;
- Polyurethane crosspolymer-1 (INCI: Polyurethane crosspolymer-1): binding;
- Polyurethane crosspolymer-2 (INCI: Polyurethane crosspolymer-2): film-forming;
- Polyamide (nylon) (INCI: Polyamide-5): improves skin condition;
- Polyamide (nylon) 6 (INCI: Nylon-6; Nylon 6/12): softening/moisturizing, improves skin condition, viscosity-regulating, filling agent;
- Polyamide (nylon) 12 (INCI: Nylon-12, Nylon-12 fluorescent brightener 230 Salt Nylon 12, Nylon 6/12): filling, darkening, viscosity-regulating;
- Styrene-acrylic copolymer (INCI: Styrene/acrylates copolymer): darkening, film-forming;
- Poly (ethylene terephthalate) (INCI: Polyethylene terephthalate): film-forming;
- Poly (ethylene isotereftalate) (INCI: Polyethylene isoterephthalate): filling, bonding, film-forming, hair fixative, viscosity-regulating, aesthetic agent;
- Poly (butylene terephthalate) (INCI: Polybutylene terephthalate): film-forming, viscosity regulating;
- Polyacrylates, acrylates copolymer (INCI: Acrylates copolymer acrylates crosspolymer): antistatic, binding, film-forming, hair fixative, suspending agent;
- Copolymer of ethylene and acrylic acid (INCI: Ethylene/acrylic acid copolymer): film-forming, gelling agent;
- Polystyrene (INCI: Polystyrene): film-forming;
- Crosspolymer of methyl methacrylate (INCI: Methyl methacrylate crosspolymer): film-forming;
- Polymethylsilsesquioxane (INCI: Polymethylsilsequioxane): darkening;
- Polylactide (INCI: Polylactic acid): abrasive.

Many cosmetic manufacturers have stopped using the polymers included in this list in their products, with the environment and consumers in mind [5].

Impact of Microplastics on the Water Environment

From year to year, the content of waste plastic particles in the seas and oceans is steadily increasing, posing quite a threat to ecosystems. The marine environment is most vulnerable to pollution of anthropogenic origin, which, together with chemical pollutants, contaminates the seabed, deep water and surface waters of all oceans, as well as beaches. Waste plastics reach the ocean with the flow of rivers, with surface and deep-sea water surges, from ships, during catastrophes, from the atmosphere and from various objects present at sea, such as oil rigs. Plastic bags and bottles, as well as fishing equipment, are the most common waste discarded by fishing boats, and this has been a huge problem for more than 50 years. Plastic waste contributes to the death of many organisms: it causes damage to their bodies, poisons the organisms by emitting harmful substances and restricts their growth by their becoming entangling in it [7,8].

As mentioned earlier, waste plastics are being fragmented into micrometer or millimeter-sized particles by:

- Ultraviolet radiation (UV) exposure;
- Waves;
- Salinity;
- Oxygen availability.

The presence of microplastics has been reported all over the world: from the intertidal zone to deep-sea sediments and from the polar regions to the equator. Millimeter-sized plastics were first noticed in the marine environment as early as the 1970s. Now, the scale of the problem is huge, as the amount of microplastics in the marine environment has been estimated to be 12.5 to 125 billion particles, according to data. Their main sources are wastewater treatment plants and overflowing drainage systems. Each year, 21% of all primary microplastics and 79% of all secondary microplastics enter marine systems. Water pollution from primary microplastics comes mainly from household chemicals (e.g., detergents and fertilizers, which are the source of the smallest plastic microbeads) and from its addition to cosmetic preparations, such as toothpastes, masks, shower gels, creams and exfoliants [3,7,9,10]. Secondary microplastics are also a significant water pollutant [3,7,10].

The presence of microplastics in the marine environment has a negative impact on fauna and flora. Its presence in marine biota is reported at all trophic levels—from phytoplankton and zooplankton to fish [11,12]. Owing to their small size and low density, they easily spread in water. The size of microplastics is a contributing factor to their ability to harm organisms—the smaller the size of the microplastic, the more often it is consumed by organisms throughout the food chain. The color of microplastics is also of great importance, as those similar in color to biological foods are more likely to be consumed, whereas the shape of microplastics affects bioaccumulation and toxicity in organisms. Depending on their shape, they can be absorbed differently. Microplastics that have accumulated in the intestines of fish can cause a range of toxic effects, which include inflammation and metabolic disorders, mucosal damage and increased mucosal permeability [13].

The generation of new environmental hazards is a result of the structure and composition of microplastics. Toxic organic pollutants found in water can be locally accumulated on the surface of microplastics, resulting in a local increase in their concentration. In addition, plastics present in the aquatic environment often release toxic compounds used in their production process (plasticizers). The biological effect of introducing toxins into the body through the ingestion of microplastics is the accumulation of them in the tissues of animals, leading to disease or even death. It is also worth noting that microplastics consumed by smaller animals are transported up the nutritional chain and can eventually end up in the body of fish consumed by humans. This means that the problem directly affects the entire human population, and not just by posing threat to the environment, as it can negatively affect the health and life of any person [11].

*1.2. Microplastics in Cosmetics*

Since the 1950s, microplastic has been added to cosmetic products in order to achieve the desired properties of the formulation. It can be found in both rinse-off cosmetics, such as shower gels, and non-rinse-off cosmetics, such as body lotions or eye shadows. Cosmetic manufacturers have favored plastic microbeads because they have proven safe to use and effective in removing dead skin cells. Compared to naturally derived ingredients, they are cheaper, less harsh, compatible with other product ingredients, wash off easily and do not cause damage to packaging containers.

In cosmetics, microspheres of different sizes and shapes are used for a variety of purposes [14]:

- In abrasive cosmetics formulations to remove dead cells of the stratum corneum;
- In toothpastes;
- In bath liquids, soaps, etc., for cleansing the skin of the body and face;
- They have decorative functions, such as glitter, in some preparations.

In facial cleansers, the microbeads used are smaller, whereas, in body scrubs, their size is larger in order to more effectively abrade the epidermis. In toothpastes, the microbeads are up to 100 times smaller, for very gentle cleansing, whereas, in face washes, they are 2 to 4 times smaller than in body scrubs [14].

Microplastics, when rinsed away, make their way into domestic sewage systems, and then to wastewater treatment plants, where large quantities of them are retained. Unfortunately, due to the small size of microplastics, many of them enter the environment. In underdeveloped countries, where there is no access to wastewater treatment plants, yet the usage of microplastics-containing formulations is very high, most of the flushed microplastics end up directly in the environment. Data from 2015 show that European Union countries, together with Switzerland, use 4130 tons of microplastics in cosmetics annually, whereas data from 2017 show that, in mainland China alone, an average of 306.9 tons of microplastics enter the aquatic environment annually [14–16].

In the current legal situation, and due to the adverse effects of microplastics on the environment and human health, it became necessary to find alternative cosmetic ingredients with abrasive potential. Therefore, the purpose of this study was to analyze the compositions of drugstore cosmetics of the "scrub" type from Polish manufacturers and identify the most common natural materials replacing microplastics.

## 2. Materials and Methods

The analysis was conducted on 50 randomly selected products from Polish manufacturers of the 'scrub' type cosmetics available in stationary drugstores in Krakow from November to December 2021. Each cosmetic was subjected to a detailed analysis of INCI (International Nomenclature of Cosmetics Ingredients) composition based on the European CosIng database and the Commission Decision of 9 February 2006 amending Decision 96/335/EC establishing an inventory and common nomenclature of ingredients used in cosmetic products. All ingredients with an abrasive function, plant extracts, vitamins, preservatives and vegetable oils were selected and categorized.

The results obtained were analyzed using Excel (Microsoft, Redmond, WA, USA) and presented in a graphical form.

## 3. Results

*Ingredients of Tested Cosmetic Products*

The presence of microplastics, and their types, in the analyzed products are shown in Figure 1. Among all analyzed abrasive cosmetics available in drugstores, microplastics were not present in 37 products. Thirteen of the remaining cosmetics contained at least one microplastic. Among the cosmetic products analyzed, one product appeared that contained two types of microplastic: INCI: Acrylates/C10-30 Alkyl Acrylate Crosspolymer and INCI: Styrene/Acrylates Copolymer.

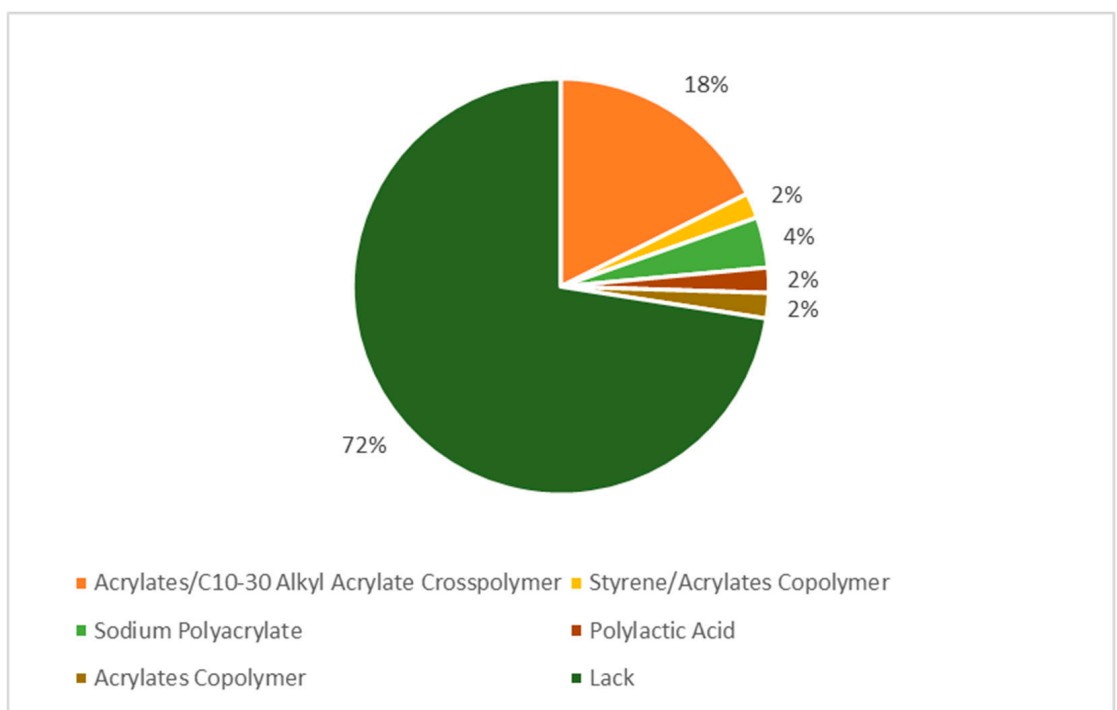

**Figure 1.** The percentage of microplastic content in the 50 cosmetic products analyzed.

The most frequently recurring microplastic was INCI: Acrylates/C10-30 Alkyl Acrylate Crosspolymer, the number of products having this ingredient was nine. INCI: Sodium Polyacrylate was the second most common microplastic and appeared in two products. INCI: Styrene/Acrylates Copolymer, INCI: Polylactic Acid and INCI: Acrylates Copolymer occurred once. Figure 2 shows a comparison of the number of plant and mineral ingredients in the analyzed cosmetics.

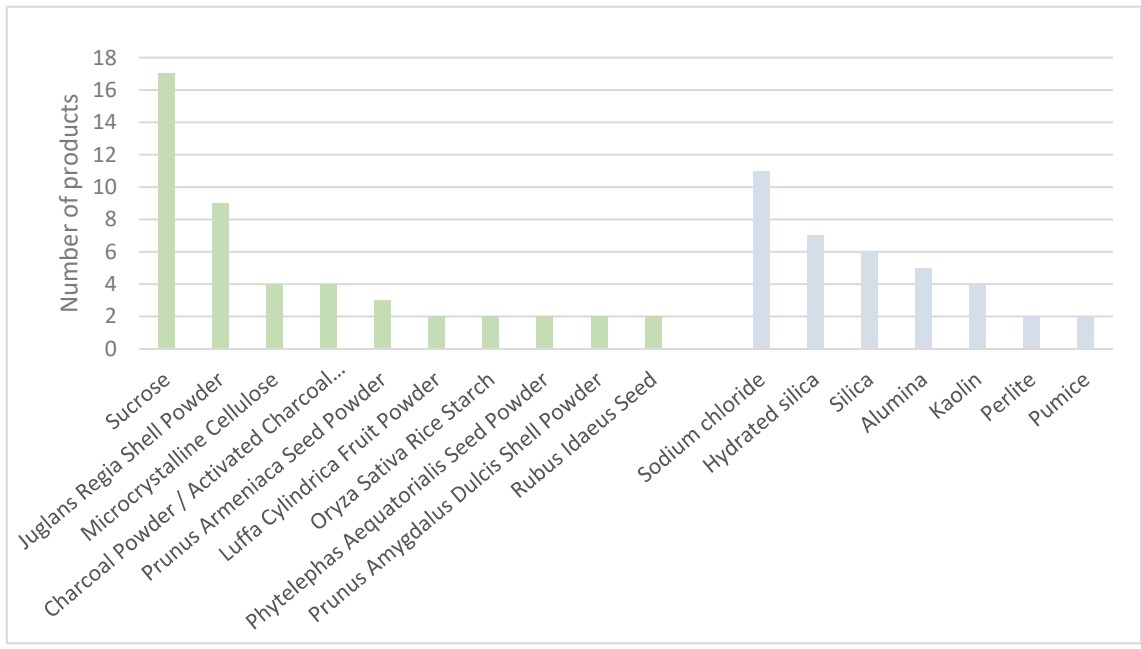

**Figure 2.** Division of abrasive ingredients taking into account the frequency of occurrence in the analyzed cosmetic products and the source of origin. Green color indicates vegetable raw materials, gray color represents mineral raw materials.

The most frequently used vegetable ingredient was sugar (INCI: Sucrose), which appeared in 34% of the products. Another frequently used ingredient was powdered walnut shells (INCI: Juglans regia Shell Powder), which appeared in 18% of cosmetics. Less than half as often used was microcrystalline cellulose (INCI: Microcrystalline Cellulose) and activated charcoal (INCI: Charcoal Powder/Activated Charcoal/Carbon Black), both of which appeared in 8% of products. Another ingredient was apricot kernel powder (INCI: Prunus Armeniaca Seed Powder), which appeared in 6% of the cosmetics tested. Other ingredients that occurred at a frequency of 4% included powdered dried Egyptian truffle fruit (INCI: Luffa Cylindrica Fruit Powder), rice starch (INCI: Oryza Sativa Rice Starch), powdered dried ground seeds of the Ecuadorian ivory palm (INCI: Phytelephas Aequatorialis Seed Powder), dried and ground almond tree fruit shell powder (INCI: Prunus Amygdalus Dulcis Shell Powder) and dried raspberry seeds (INCI: Rubus Idaeus Seed).

The most commonly used mineral ingredient was sodium chloride (INCI: Sodium Chloride), which appeared in 22% of the cosmetic formulations. Another mineral ingredient was silicic acid (INCI: Hydrated Silica), which appeared in 14% of the studied formulations. The next ingredient, silica (INCI: Silica), occurred in 12%. Alumina (INCI: Alumina) was identified in 10%, and white clay (INCI: Kaolin) in 8% of cosmetics. Other ingredients, i.e., perlite (INCI: Perlite) and pumice (INCI: Pumice), appeared in 4% of the tested formulations.

The analysis of cosmetic products also included verification of the presence frequency of plant extracts, vitamins, preservatives and plant oils, as shown in the following Tables 1–4.

Table 1 shows the frequency of plant extracts. Of the 50 products analyzed, 16 did not contain any extract. The most frequently repeated ingredients were kiwi water extract (INCI: Actinidia Chinensis (Kiwi) Fruit Extract) and lemon extract (INCI: Citrus Limon Fruit Extract). Other plant extracts occurred only once in all analyzed cosmetics.

Table 2 summarizes the vitamins found in the analyzed cosmetic formulations. Vitamin E and its derivatives were found in 29 formulations, various forms of vitamin C in 7, and panthenol in 6.

Among all the cosmetic products that were analyzed, no preservative was found in 10 of them, whereas the most frequently recurring one was sodium benzoate (INCI: Sodium Benzoate). Potassium sorbate (INCI: Potassium Sorbate), phenoxyethanol (INCI: Phenoxyethanol) and benzyl alcohol (INCI: Benzyl Alcohol) were also frequently present. The results are summarized in Table 3.

**Table 1.** Frequency of plant extracts in the analyzed abrasive cosmetic products.

| Plant Extracts | | Frequency per 50 Products |
|---|---|---|
| **Ingredient** | **INCI Name** | |
| None | - | 16 |
| Kiwi Water Extract | Actinidia Chinensis Fruit Extract | 4 |
| Lemon Extract | Citrus Limon Fruit Extract | 4 |
| Bamboo Extract | Bambusa Arundinacea Stem Extract | 2 |
| Bamboo Shoot Extract | Bambusa Vulgaris Shoot Extract | 2 |
| Fig Fruit Extract | Ficus Carica (Fig) Fruit Extract | 2 |
| Blueberry Fruit Extract | Vaccinium Myrtillus Fruit Extract | 2 |
| Other Plant Extracts | | 43 |

**Table 2.** Frequency of vitamins in the analyzed abrasive cosmetics products.

| | INCI Name | Frequency per 50 Products |
|---|---|---|
| None | - | 19 |
| Vitamin E Group | Tocopherol | 17 |
| | Tocopheryl Acetate | 10 |
| | Tocopherol (Mixed) | 1 |
| | Tocopheryl Stearate | 1 |
| Vitamin C Group | Ascorbyl Palmitate | 4 |
| | Ascorbic Acid | 2 |
| | Ascorbyl Tetraisopalmitate | 1 |
| Vitamin B Group | Panthenol | 6 |

**Table 3.** Frequency of preservatives in analyzed abrasive cosmetics.

| INCI Name | Frequency per 50 Products |
|---|---|
| Sodium Benzoate | 22 |
| Potassium Sorbate | 17 |
| Phenoxyethanol | 16 |
| Benzyl Alcohol | 12 |
| None | 10 |
| Dehydroacetic Acid | 9 |
| Benzoic Acid | 6 |
| DMDM Hydantoin | 4 |
| Methylparaben | 3 |
| Propylparaben | 3 |
| Sodium Salicylate | 3 |
| Other | 7 |

**Table 4.** Frequency of vegetable oils in analyzed cosmetic products.

| Vegetable Oils | | Frequency per 50 Products |
|---|---|---|
| **Ingredient** | **INCI Name** | |
| None | - | 16 |
| Coconut Oil | Cocos Nucifera Oil | 9 |
| Soybean Oil | Glycine Soja Oil | 8 |
| Sunflower Seed Oil | Helianthus Annuus Seed Oil | 8 |
| Sweet Almond Oil | Purnus Amygdalus Dulcis Oil | 8 |
| Grapeseed Oil | Vitis Vinifera Seed Oil | 8 |
| Macadamia Nut Oil/ Macadamia Seed Oil | Macadamia Ternifolia Seed Oil/ Macadamia Integrifolia Seed Oil | 7 |
| Argan Oil | Argania Spinosa Kernel Oil | 3 |
| Sweet Orange Peel Oil | Citrus Aurantium Dulcis Peel Oil | 3 |

**Table 4.** *Cont.*

| Vegetable Oils | | Frequency per 50 Products |
| --- | --- | --- |
| **Ingredient** | **INCI Name** | |
| Hydrogenated Castor Oil | Hydrogenated Castor Oil | 3 |
| Olive Oil | Olea Europaea Fruit Oil | 3 |
| Oil palm Oil | Elaeis Guineensis (Palm) Oil | 2 |
| Cotton seed Oil | Gossypium Herbaceum Seed Oil | 2 |
| Vegetable Oil | Olus Oil | 2 |
| Avocado Oil | Persea Gratissima Oil | 2 |
| Raspberry Seed Oil | Rubus Idaeus (Raspberry) Seed Oil | 2 |
| Other Oils | | 35 |

No vegetable oil appeared in the 16 analyzed products. Of all the oils, the most commonly used were coconut oil (INCI: Cocos Nucifera Oil) and soybean oil (INCI: Glycine Soja Oil). Other vegetable oils were found only once in all analyzed cosmetics. The frequency of occurrence of each oil in the formulations of the studied cosmetics is summarized in Table 4.

## 4. Discussion

The stratum corneum is the outermost layer of the epidermis, where the final stage of keratinocyte maturation and development occurs. In the basal layer of the epidermis, keratinocytes are proliferating, but as they mature, they lose their proliferative potential and undergo programmed cell destruction. The final form of keratinocytes are corneocytes, which are differentiated, nucleus-free cells with preserved keratin fibers. The stratum corneum consists of about 15 layers of flattened corneocytes and is divided into two parts: the compact layer (stratum compactum) and the keratinizing layer (stratum disjuntcum) [17].

Removal of dead skin is carried out with chemical and mechanical peels or by microdermabrasion treatment. A safe and very simple way to do this at home is by mechanical peeling. This requires a cosmetic product with abrasive grains of different shape, hardness and size, which determines the exfoliating power of the peel [18]. As indicated above, these grains could be obtained from natural (mineral or plant) sources or extracted synthetically. The second treatment method has been used remarkably often, due to its low cost, durability of the material and lack of formulation incompatibilities with other ingredients in cosmetic formulations. However, just as the durability of polymeric abrasive ingredients in a cosmetic formulation accounted for their benefits, once they entered the environment, they showed their other, harmful side.

The cosmetics industry is one of the most important sources of plastic waste. In this regard, in 2020, Piotrowska et al. [19] performed the analysis of professional and drugstore cosmetics of the "scrub" type available on the Polish market. They were evaluated in terms of the content of microplastics and natural abrasive particles. The proportional shares of preparations containing polyethylene, ingredients of natural origin or of mixed composition were also determined. The results showed that, of the 130 cosmetics analyzed, 58 (44.6%) were products with a natural composition, 50 (38.5%) were products containing polyethylene, and 22 (17%) had a mixed composition, containing polyethylene as well as abrasives of natural origin. Every product intended for professional use contained polyethylene in its formulation. Among the 22 cosmetics with mixed composition, 17 contained one type of natural abrasive, 4 products contained two types, and 1 product contained three ingredients of natural origin.

The place of microplastics is increasingly beginning to be supplanted by natural ingredients with abrasive capacity. Piotrowska et al. [19], in a study conducted in 2017, showed that, out of 130 cosmetics, 80 contained natural abrasive particles, where the most common natural exfoliating ingredients in cosmetics containing polyethylene were crushed seeds and microcrystalline cellulose, whereas seeds, sugar and salt were the most common such ingredients in cosmetics with natural ingredients. In the present study, natural abrasive particles appeared in 49 of the cosmetics analyzed, but the most common ingredients were sugar, sodium chloride and powdered walnut shells. Sugar was the most common vegetable substitute for microplastic, whereas sodium chloride was the most common mineral substitute. Only one formulation did not contain any natural ingredient with an abrasive effect [19].

Very few studies contain analysis of purifying products containing microbeads in the formulation, for the purpose of identification. The microsphere content of the final product can range from 0.05% to 12%, so it is impossible to accurately calculate the average content in cosmetic products. Nevertheless, new studies have shown a significantly lower percentage of microspheres in peeling and cleansing products [15,20]. These values are expected to gradually decrease as most companies voluntarily, or under legal pressure, begin to phase them out. In the present study, out of 50 cosmetics analyzed, 37 did not contain any type of microplastic. Only 13 products contained it (of which one formulation contained two types of microplastic), and the most common was Acrylates/C10-30 Alkyl Acrylate Crosspolymer. Most of the cosmetics tested were characterized by an all-natural composition. This is a big change from previous studies [15,19,20].

The cost of producing synthetic polymers is lower than the cost of obtaining good quality natural raw materials. Such raw materials must be obtained from the natural resources, processed, extracted and stored under appropriate conditions in order to avoid decomposition or growth of microorganisms. Synthetic products are clean and more resistant to decay or contamination. In addition, various analyses indicate that the production costs of biopolymers are high [21], as is the production of easily degradable bioplastics [22]. Therefore, synthetic polymers are used much more frequently by cosmetics manufacturers.

Cosmetic products are very often enriched with plant oils or extracts, as well as vitamins. Vegetable oils and extracts did not appear in only 16 products; the most common oil was coconut oil and the most common extracts were kiwi water extract and lemon extract.

Polyethylene is the most common synthetic material produced in the world. Its molecules, weighing more than 1000 daltons, are considered completely non-degradable. The ECHA, in 2019, put forward a proposal to restrict the use of 19 polymeric plastic compounds in cosmetic products. Piotrowska et al. showed that 72 products contained polyethylene as an exfoliating raw material. In the present study, polyethylene did not appear in any of the analyzed abrasive cosmetics. This shows that the Polish cosmetic industry complies with the legal requirements of the European Union and follows the recommendations for the ban on the addition of polyethylene abrasive ingredients, which was introduced in 2019. Based on this, it is easy to conclude that many manufacturers have stopped using this ingredient, having been motivated by the welfare of the consumer and the environment [15,19].

Pressure from NGOs, such as the Plastic Soup Foundation, media campaigns and abundant scientific evidence of the adverse effects of microplastics on aquatic organisms have prompted intergovernmental organizations, namely the United Nations (UN) and the G7 (Group of Seven), to adopt plans to reduce microplastic waste in marine environments.

In the US, the state of Illinois was the first to ban non-biodegradable microbeads in personal care products in 2014, followed by 15 other states in the US. A year later, the US federal law, the Microbead-Free Waters Act, was voted on; this act bans the production and sale of intentionally added non-biodegradable plastic microspheres in rinse-off products as of July 2017, and as of July 2018, excludes biodegradable plastic microbeads. The decision had an immediate impact on other countries that were hesitant to take action. In 2019,

China issued new guidelines for Chinese industry to completely ban the sale of household chemicals containing microbeads from 31 December 2022 [14].

In the European Union, the Dutch government was the first official body to suggest a ban on microbeads, in 2013, and emphasized the need to inform the public about their environmental impact. Scandinavian countries have been pioneers in promoting legislation against plastic microbeads in cosmetics [14].

In 2017, at the request of the European Commission, Amec Foster Wheeler, a consulting firm, conducted a study on intentionally added plastic microplastics in the EU. As a result of the findings, the European Commission charged the ECHA with preparing a possible restriction on the intentional addition of synthetic, insoluble microplastics to cosmetics. In 2019, the ECHA proposed definitions for a series of polymers and their use, as well as an EU-wide ban on certain products containing microplastic substances or mixtures, in order to reduce emissions across the EU. This has resulted in most member states limiting themselves to plastic microbeads in exfoliating cosmetics and cleansing products. In June 2020, the RAC (Risk Assessment Committee) and SEAC ECHA (Socio-Economic Analysis Committee) draft committees were combined and proposed a ban on microplastics above 1 μm and up to 5 mm, thereby increasing the lower diameter from 1 nm—as in the definition of microplastic—to 100 nm. The first country in the European Union to follow REACH's (Registration, Evaluation and Authorization of Chemicals) proposed definitions of plastics, microplastics and nanoplastics with a national ban on microplastics in 2019 was Ireland [14].

There are still many legislative errors: although all active bans forbid the sale of cosmetics with microspheres, there is no clear indication that the production and import of such products is also restricted. An example of this is Canada, which has banned the manufacture and addition of microspheres to Canadian cosmetics, but still allows them to be transported through the country.

Statistics presented by Cosmetics Europe, show that there has been a 97.6% decrease in microspheres since 2017, but this are not scientifically supported data. The declining number of microbeads, as a result of their replacement with natural alternatives, following the announcement of the above statistics, has caused worldwide satisfaction, and at the same time, a reluctance on the part of governments to impose restrictions, as they consider them unnecessary in this situation.

As of now, microbeads are still allowed in many countries. Few European countries have imposed restrictions. The European Chemicals Agency, at the end of 2022, plans to introduce EU regulations restricting the intentional addition of microplastics, which will certainly change the Polish cosmetics market, and manufacturers will face new challenges [21].

## 5. Conclusions

Among the 50 cosmetics analyzed, 49 contained natural abrasive particles. Thirteen abrasive cosmetics contained microplastics in their composition. The most common microplastic was INCI: Acrylates/C10-30 Alkyl Acrylate Crosspolymer, whereas sugar was the most common vegetable substitute for microplastic, and sodium chloride was the most common mineral substitute. Polyethylene did not occur in any of the cosmetics analyzed. Analyzing the composition of scrub-type preparations, in comparison with previous years, one can see a positive change in the Polish cosmetic market.

**Author Contributions:** Conceptualization, A.P.; methodology, A.P.; software, O.C.-L.; validation, A.P., M.G. and O.C.-L.; formal analysis, M.G.; investigation, M.G.; resources, A.P.; data curation, M.G.; writing—original draft preparation, M.G.; writing—review and editing, A.P. and O.C.-L.; visualization, A.P., M.G. and O.C.-L.; supervision, A.P.; project administration, A.P. All authors have read and agreed to the published version of the manuscript.

**Funding:** This research received no external funding.

**Institutional Review Board Statement:** Not applicable.

**Informed Consent Statement:** Not applicable.

**Data Availability Statement:** The data presented in this study are available on request from the corresponding author.

**Conflicts of Interest:** The authors declare no conflict of interest.

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
