# Peer review of "Compositions of Abrasive Cosmetics from Polish Manufacturers"

_cosmetics, doi:10.3390/cosmetics10020067_

Round 1

Reviewer 1 Report

The study focuses on to analyse the compositions of widely available rinse-off abrasives cosmetics from Polish manufacturers and to identify the most common natural raw materials replacing microplastics. The scientist has analysed fifty randomly selected rinse-off products for abrasive ingredients in International Nomenclature of Cosmetics Ingredients formulation produce by Polish manufactures. The topic is interesting and having effect on both environmental as well as human on long term. The minor observations are as follows:

1.       Line 203: Heading 3.1, should be corrected to: 3.1. Ingredients of tested in cosmetic products

2.       The clarity of figure 1 low. Need to improve.

3.       Figure 3.1: Not able to differentiate, Acrylate copolymers and Lack  

4.       In discussion part of result, don’t only describe results/findings in number, please corelated why particular microplastic, natural ingredients, plant extract, preservative etc used by that manufacturing, cost or they are effective abrasive agent/preservative/flavouring agent etc.

5.       In table 1 Last raw. What is Other 43?

6.       In table 4 Last raw: What is Other 35?

Author Response

Dear Reviewer,

We would like to thank you for the time you devoted to reading and reviewing our manuscript. Your valuable comments have helped to improve the quality of our work.

Best regards,

Authors

The study focuses on to analyse the compositions of widely available rinse-off abrasives cosmetics from Polish manufacturers and to identify the most common natural raw materials replacing microplastics. The scientist has analysed fifty randomly selected rinse-off products for abrasive ingredients in International Nomenclature of Cosmetics Ingredients formulation produce by Polish manufactures. The topic is interesting and having effect on both environmental as well as human on long term. The minor observations are as follows:

  1. Line 203: Heading 3.1, should be corrected to: 3.1. Ingredients of tested in cosmetic products

Thank you for noticing this editing oversight – it has been corrected.

  1. The clarity of figure 1 low. Need to improve.

Figure 1 has been inserted in better quality

  1. Figure 3.1: Not able to differentiate, Acrylate copolymers and Lack  

Figure has been corrected

  1. In discussion part of result, don’t only describe results/findings in number, please corelated why particular microplastic, natural ingredients, plant extract, preservative etc used by that manufacturing, cost or they are effective abrasive agent/preservative/flavouring agent etc.

Thank you for this important remark. A paragraph on the topic of production costs has been added to the discussion.

  1. In table 1 Last raw. What is Other 43?

Data corrected in table and description added in text.

  1. In table 4 Last raw: What is Other 35?

Data corrected in table and description added in text.

Reviewer 2 Report

The article "Compositions of abrasives cosmetics from Polish manufacturers", from my point of view, is very interesting and brings back to everyone's attention how important it is to keep Planet Earth clean and uncontaminated.

The material proposed by the collective of authors can also be considered an alarm signal regarding the simple and easy way of circulation in nature of plastic nanoparticles and the harmful effects induced at all levels by these contaminants.

Starting from the regulations in force regarding the monitoring of the presence of different chemical contaminants, the article aims to evaluate an appreciable number of cosmetic products on the pharmaceutical or professional market in Poland.

Based on the evaluation of the article, we would find:

- The introduction is judiciously written, it provides information about plastic nanoparticles, about their origin, the harmfulness induced to the environment and to humans upon repeated exposure, the types of substances used in dermatocosmetics, especially as abrasive material, as well as information about the directions that certain manufacturers of dermatocosmetics follow them precisely in order to limit and propagate the effects induced by these types of contaminants

- The Material and Methods section contains information on all types of protocols applied in the analysis of 50 randomly selected dermatocosmetic products, all produced in Poland; the authors mention that each formula has been judiciously analyzed from the point of view of composition

- The results are based on the information obtained by the authors following the analysis of dermatocosmetic formulations; unfortunately in this cosmetics market, the authors sound the alarm on the most used ingredient Acrylates/C10-30 Alkyl Acrylate Crosspolymer and INCI: Styrene/Acrylates Copolymer; to express these results, the authors use value data entered in tables but also very justifiable graphs; the authors also present the alternative methods used by certain manufacturers in order to limit the use of plastic nanoparticles, in this direction the types of vegetable extracts and their derivatives used, the types of vitamins, vegetable oils and the nature of preservatives are presented

- The discussions are clearly presented, and the data obtained from the theoretical analysis of dermatocosmetic formulations are reported to the specialized literature; all the aspects presented by the authors in this sub-chapter are harmonized with the legislative provisions at the European level and not only

- The conclusions are clear and succinctly point out the purpose and objectives of this study, and the dermatocosmetics market in Poland seems to have understood very well the limitation of certain types of contaminants and has focused a lot on bio natural products of plant origin in ensuring the induction of the desired effects, that we are talking about cleansing, peeling lotions, etc.

- The bibliography is consistent with the data presented in the article.

In conclusion, I agree with the publication of the article.

Author Response

Dear Reviewer,

We would like to thank you for the time you devoted to reading and reviewing our manuscript. We are very pleased to have such an appreciative review of our work.

Best regards,

Authors